A metagenomic comparison of clearwater, probiotic, and Rapid BFTTM on Pacific whiteleg shrimp, Litopenaeus vannamei cultures

http://orcid.org/0000-0002-2632-3612 Waiho Khor 1 2 3 waihokhor@gmail.com
Abd Razak Muhammad Syafiq 1 4
Abdul Rahman Mohd Zaidy 4
Zaid Zainah 4
Ikhwanuddin Mhd 1 3 5
Fazhan Hanafiah 1 2 3
http://orcid.org/0000-0003-3014-442X Shu-Chien Alexander Chong 2 6
http://orcid.org/0000-0002-0708-5970 Lau Nyok-Sean 2
http://orcid.org/0000-0002-9510-224X Azmie Ghazali 1
Ishak Ahmad Najmi 1
http://orcid.org/0000-0002-3823-7535 Syahnon Mohammad 1 7
http://orcid.org/0000-0002-3041-199X Kasan Nor Azman 1 3 norazman@umt.edu.my
1 Higher Institution Centre of Excellence (HICoE), Institute of Tropical Aquaculture and Fisheries, Universiti Malaysia Terengganu , Kuala Nerus , Malaysia
2 Centre for Chemical Biology, Universiti Sains Malaysia , Minden, Penang , Malaysia
3 STU-UMT Joint Shellfish Research Laboratory, Shantou University , Shantou, Guangdong , China
4 Zaiyadal Aquaculture Sdn. Bhd. , Shah Alam, Selangor , Malaysia
5 Faculty of Fisheries and Marine, Universitas Airlangga , Surabaya , Indonesia
6 School of Biological Sciences, Universiti Sains Malaysia , Minden, Penang , Malaysia
7 Centre of Research and Field Service (CRaFS), Universiti Malaysia Terengganu , Kuala Nerus, Terengganu , Malaysia
Esteban María Ángeles
Electronic publication date: 2023 Sep 28
Publication date: 2023
Volume: 11
Electronic Location ID: e15758
Received 2023 Mar 23; Accepted 2023 Jun 26
Copyright: © 2023 Waiho et al.
Copyright year: 2023
Copyright holder: Waiho et al.
License: This is an open access article distributed under the terms of the Creative Commons Attribution License, which permits unrestricted use, distribution, reproduction and adaptation in any medium and for any purpose provided that it is properly attributed. For attribution, the original author(s), title, publication source (PeerJ) and either DOI or URL of the article must be cited.
License URL: https://creativecommons.org/licenses/by/4.0/

Keywords: Rapid BFTTM, Penaeus vannamei, Biofloc, Probiotic, Metagenome

Funding: This work was supported by the Ministry of Higher Education, Malaysia under the Higher Institution Centre of Excellence (HICoE) Program accredited to Institute of Tropical Aquaculture and Fisheries, Universiti Malaysia Terengganu [Vot no: 63933 & Vot no: 56050] and the Knowledge Transfer Programme (KTP)—Lifelong Learning Scheme grant [LLR(R5)-FS/9(UMT-20)] awarded to the authors and their collaborative industry partner, Zaiyadal Aquaculture Sdn. Bhd. The funders had no role in study design, data collection and analysis, decision to publish, or preparation of the manuscript.

==============================
Biofloc technology improves water quality and promote the growth of beneficial bacteria community in shrimp culture. However, little is known about the bacteria community structure in both water and gut of cultured organisms. To address this, the current study characterised the metagenomes derived from water and shrimp intestine samples of novel Rapid BFTTM with probiotic and clearwater treatments using 16S V4 region and full length 16S sequencing. Bacteria diversity of water and intestine samples of Rapid BFTTM and probiotic treatments were similar. Based on the 16S V4 region, water samples of >20 μm biofloc had the highest abundance of amplicon sequence variant (ASV). However, based on full length 16S, no clear distinction in microbial diversity was observed between water samples and intestine samples. Proteobacteria was the most abundant taxon in all samples based on both 16S V4 and full length 16S sequences. Vibrio was among the highest genus based on 16S V4 region but only full length 16S was able to discern up to species level, with three Vibrios identified—V. harveyi, V. parahaemolyticus and V. vulnificus. Vibrio harveyi being the most abundant species in all treatments. Among water samples, biofloc water samples had the lowest abundance of all three Vibrios, with V. vulnificus was present only in bioflocs of <20 μm. Predicted functional profiles of treatments support the beneficial impacts of probiotic and biofloc inclusion into shrimp culture system. This study highlights the potential displacement of opportunistic pathogens by the usage of biofloc technology (Rapid BFTTM) in shrimp culture.

Key points

Rapid BFTTM and probiotic treatments had similar bacterial communities.

Proteobacteria was the most abundant taxon across treatments.

Bioflocs (<20 μm) had more candidate pathogens.

Introduction

Whiteleg shrimp, Penaeus vannamei is among the world most valuable crustacean species, with its global production increased approximately 143% within a short period of 5 years and accounted for 5.46 million tonnes in 2019. Furthermore, almost 80% of the global P. vannamei production was from Asia (FAO, 2021). The high global demand for P. vannamei has pushed forward its intensification process and now P. vannamei can be cultured super-intensively (Khoa et al., 2020). Super-intensive culture of any aquatic organisms, including P. vannamei, could easily lead to unwanted stress due to reduced water quality and limited open space (Waiho et al., 2021) that could subsequently affect growth (Esparza-Leal et al., 2020) and innate immunity (Shinji et al., 2019), and result in mortality and lower economic output.

Traditionally, chemicals and drugs such as antibiotics are being used to control the prevalence of diseases in super intensive aquaculture system of P. vannamei. However, aside from the hazardous residual problems in the final product (P. vannamei) that is a major concern of food safety (Chinabut & Puttinaowarat, 2005; Dada et al., 2021), excessive antibiotic residues have also led to the development of antibiotic resistant bacteria found in farmed P. vannamei (Thornber et al., 2020). Although other methods such as the use of plant-based compounds as alternatives are being explored (Wu et al., 2021), commercial P. vannamei farms are starting to incorporate biofloc technology systems (BFT) into their cultures in replacement of antibiotics (Arias-Moscoso et al., 2018). BFT is being regarded as sustainable and green as it reduced the need for water exchange (up to zero water exchange is feasible) while still maintaining optimum water quality in super high intensity aquaculture systems of fish and crustacean species, including P. vannamei (Ahmad et al., 2017; Kasan et al., 2021a). By optimising the carbon:nitrogen (C:N) ratio, microbial community, especially heterotrophic bacteria, most of which have probiotic characteristics, proliferates in the presence of accumulated organic matter and nutrients within the optimum quantity. The highly diverse pool of microorganisms plays critical role in stabilising the overall water column within the culture system, from maintaining water quality by converting harmful nitrogenous compounds into in situ microbial protein to promote the health of cultured organisms by increasing the percentage of beneficial microorganisms within the water and intestines (Abakari, Luo & Kombat, 2021; El-Sayed, 2021). However, since bioflocs are an amalgamation of various biological and non-biological components, including bacteria, phytoplankton, zooplankton, uneaten feed, feces, etc., Romano (2021), the probiotics found within bioflocs are largely unknown.

In addition to BFT, commercial super-intensive P. vannamei cultures have started to use commercially live microbial strains with well-established impacts on hosts known as probiotics. Probiotics such as lactic acid bacteria can be introduced either via diet or directly administered into the water column of shrimp culture systems. Being considered as bio-friendly, probiotics are non-pathogenic and do not exert adverse effect towards the cultured organisms. Instead, they regulate and compete with pathogenic bacteria, consequently resulting in the positive growth of aquatic organisms (Farzanfar, 2006). For example, Lactococcus lactis subsp. lactis was isolated from the intestine of healthy P. vannamei and exhibited probiotic effect for P. vannamei due to its ability to inhibit the growth of Vibrio anguillarum and V. harveyi (Adel et al., 2017).

As the underlying mechanisms of BFT remains unknown, it is essential to investigate the changes in bacterial community of cultured water column and the intestines of cultured organisms, as well as to compare with other beneficial bacteria-inducing method, i.e., probiotics (Liu et al., 2019). To achieve this, three treatments were used to culture P. vannamei in commercial farm, namely Rapid BFTTM (RBFT)—a biofloc-inducing inoculation developed by Kasan NA (Kasan et al., 2021a), commercial probiotic, and clearwater. The microbial community of water and intestines of P. vannamei were further characterised by targeting the shorter bacterial 16S V4 region or full 16S sequence.

Methodology

Shrimp culture and sample collection

Indoor shrimp culture was conducted at a commercial shrimp farm in Perak, Malaysia. Shrimps were divided into three treatments, (1) RBFT (1 × 109 colony forming unit per ml, CFU/ml) (Universiti Malaysia Terengganu (UMT), Terengganu, Malaysia; UMT Patent ID: PI 2017703679; Trademark ID: TM2021014913); (2) Commercial probiotic-containing Bacillus sp. (probiotic; ShrimpShield™) (1 × 109 CFU/ml) (Keeton Industries, Wellington, Colorado, USA); and (3) clearwater as control. Each treatment consisted of three circular tanks, each with a volume of 12 m3. Water was dechlorinated with calcium hypochlorite at 30 ppm prior usage. Experiment conducted in this study complied with the ARRIVE guidelines and carried out in accordance with the U.K. Animals (Scientific Procedures) Act, 1986 and associated guidelines, EU Directive 2010/63/EU for animal experiments. Approximately 5,000 P. vannamei post-larvae (PL) stage 21 (super juvenile) were stocked into each tank with a stocking density of 400 ind./m3. Daily water exchange of 40% was implemented. Feeding was conducted four times per day and the amount varied depending on the feed left on the feeding tray. The application of calcium hypochlorite at a concentration of 30 ppm, followed by rigorous aeration was conducted at the beginning of the experiment to ensure complete chlorine removal. Aeration was facilitated through the facility’s central system, employing aeration tubing to each tank to maintain dissolved oxygen levels above 5 ppm. Daily monitoring of dissolved oxygen was performed using a YSI multiparameter system (Model 13M10065, USA). To ensure adequate mineral content, the water was supplemented with calcium hydroxide to maintain a concentration of at least 200 mg/L of calcium and with magnesium chloride to maintain a concentration of at least 800 mg/L of magnesium. Calcium hydroxide was also used to sustain an alkalinity level of over 150 mg/L and a pH above 7.5. The daily assessment of sedimentable solids in the biofloc system was conducted using an Imhoff cone. Weekly measurements of total ammonia nitrogen (TAN), nitrite (NO2−), and phosphate were performed using a UV-vis1800 Shimadzu spectrophotometer.

RBFT and probiotic inoculums were added into the cultured water at 4 ppm (4 × 103 CFU/ml; 4.8 × 1010 CFU/tank) weekly. Molasses was introduced daily into RBFT and probiotic treatments (50% of the daily feed weight) to maintain a C:N ratio of 15:1. Water exchange was reduced to 10% daily when biofloc sedimentable solid level was above 20 ml/l (Kasan et al., 2021b). A 70-day experimental duration was implemented across treatments. Survival rate and biomass per volume were calculated on day 70.

On the last day of culture, five shrimps from each replicate tank were dissected and their pooled intestines were collected and stored in 95% ethanol solution. Water samples (500 ml per replicate, triplicates were collected from each tank and pooled together) from each treatment were also collected on the last day of shrimp culture. The water samples of RBFT treatment were further divided into two categories, <20 and >20 μm. Bacteria sample from the water samples were filtered using Whatman filter paper (no. 4) and the residues left on the filter paper (considered as size >20 μm) were stored in 95% ethanol. The filtrates (<20 μm) were then centrifuged at 10,000 rpm for 5 min. Pellet was collected and stored in 95% ethanol solution (<20 μm samples) for subsequent analyses.

DNA extraction

Water and intestine samples of each treatment (clearwater, probiotic and Rapid BFTTM) were subjected to DNA extraction. Samples were removed from 95% ethanol solution and rinsed with double distilled water. Subsequently, samples were homogenized in CTAB lysis buffer containing 0.1 and 0.5 mm silica bead using a TacoPrep bead Beater (GeneReach Biotechnology Corp., Taiwan) for 10 min. The homogenate was incubated at 65 °C for 30 min before equal volume of chloroform was added. After brief inversion to ensure thorough mixing, the homogenate was centrifuged at 10,000 × g for 5 min to allow for phase separation. The upper layer was transferred to new tube before the addition of 0.7 volume of isopropanol and 15 μl of SPRI magnetic beads. The mixture was left at room temperature for 10 min with gentle mixing to allow DNA-magnetic bead binding. The DNA-bound magnet was washed twice with 70% ethanol and DNA was subsequently eluted using 100 μl of 0.1 × TE buffer. Each treatment consisted of three replicates for the sequencing of 16S rRNA V4 gene region, and duplicates for the sequencing of full length 16S rRNA gene.

Library preparation and sequencing

Illumina iSeq100

Amplification of the microbial 16S rRNA V4 gene region from the extracted gDNA was performed using OneTaq 2x Master Mix (NEB, Ipswich, MA, USA). Primer pair 515F-806R containing partial Illumina Nextera adapter in their 5′ end was used to amplify the 16S V4 region (Walters et al., 2015). The polymerase chain reaction (PCR) amplification profile was as follows; Initial denaturation: 94 °C for 30 s, followed by 35 cycles of denaturation (94 °C for 15 s), annealing (48 °C for 15 s) and extension (68 °C for 30 s). The PCR products were bead-purified and subsequently indexed to incorporate dual-index barcode and Illumina adapter. The indexed PCR products were pooled, purified, and measured using Denovix high-sensitivity fluorescence quantification kit (Denovix, Wilmington, DE, USA). Single-end sequencing (300 bp × 1) was performed on Illumina iSeq 100 platform (Illumina, San Diego, CA, USA) by GeneSEQ Sdn. Bhd. (Malaysia).

Nanopore

Amplification of the full length 16S rRNA gene was performed using OneTaq 2X Master Mix. Universal 16S rRNA primers AGAGTTTGATYMTGGCTCAG (forward) and TACGGYTACCTTGTTACGACTT (reverse) containing Nanopore adapter at their 5′ end was used (Frank et al., 2008). The PCR amplification profile was as follows; Initial denaturation: 94 °C for 30 s, followed by 35 cycles of denaturation (94 °C for 15 s), annealing (50 °C for 15 s) and extension (68 °C for 90 s). The PCR products were diluted (1:10) and used as template for index PCR reaction to incorporate Nanopore barcode. The indexed PCR products were then pooled, gel-extracted, and measured using Denovix high-sensitivity fluorescence quantification kit. Sequencing was performed using a Minion nanopore sequencer (Nanopore Technologies, Oxford, UK).

Data analysis

16S rRNA V4 region

Non-biological forward and reverse primer sequences were removed from the raw single-ed demultiplexed reads using Cutadapt v1.18 Program (Martin, 2011). Low quality bases with a Phred score of less than 20, and reads with an unexpected error rate of 1% or higher were also trimmed using Cutadapt v1.18. The trimmed reads were used to generate amplicon sequence variant (ASV) and abundance table using DADA2 (Callahan et al., 2016) within the QIIME2 v2020.8 pipeline (Bolyen et al., 2019). The QIIME2 scikit-learn naïve Bayes machine-learning classifier, q2-feature-classifier plugin (Bokulich et al., 2018), was used for taxonomic assignment of ASVs based on the Genome Taxonomy Database r95 comprised of 191,527 bacterial and 3,073 archaeal genomes (Kaehler et al., 2019; Parks et al., 2020). The default balanced parameters for uniform weights were used, as recommended by Bokulich et al. (2018). Non-mitochondrial and non-chloroplast ASVs that were classified at least to the phylum level were used to construct an ASV abundance table. The filtered abundance table, taxonomic assignment output, and sample metadata were analysed on the MicrobiomeAnalystCA webserver (Chong et al., 2020). A low count filter (minimum count of four, and at least 20% prevalence in samples) was applied. Data were scaled using total sum scaling (TSS) normalisation factor to minimise technical bias due to variation in sequencing depth between libraries. Beta diversity profiling using Non-metric Multi-dimensional Scaling (NMDS) was conducted based on Bray-Curtix dissimilarity index and analysed using permutational MANOVA (PERMANOVA) on the number and relative abundance of ASVs. Alpha diversity within samples was estimated with ASV number and Shannon biodiversity index. Difference in alpha diversity indexes was analysed using one-way analysis of variance (ANOVA) and significant differences were highlighted using post hoc Tukey’s test. The linear discriminant analysis (LDA) effect size (LefSe) algorithm (Segata et al., 2011) in MicrobiomeAnalystCA was employed to evaluate differential abundant features, with FDR-adjusted P-value cutoff being set at 0.1 and Log LDA score at 2.0. Pearson’s correlation was used to identify any linear relationships exist between two taxa. The metagenomic functional analysis were predicted using the software PICRUSt2 based on 16S rRNA ASVs. PICRUSt2 (v2.5.0) was used with default settings to infer the MetaCyc pathway functions. LEfSe was used to analyse differentially abundant predicted MetaCyc functions by applying linear discriminant analysis (LDA) score of 3.0.

Full length 16S rRNA

Basecalling of raw nanopore reads was performed using Guppy Basecalling Software version 4.4.0 (Nanopore Technologies, Oxford, UK) at high accuracy mode and demultiplexed using Guppy Barcoder at default settings. NanoClust 16S rRNA analysis pipeline (Rodríguez-Pérez, Ciuffreda & Flores, 2020) was used to perform UMAP-based classification based on the demultiplexed full length 16S rRNA nanopore reads. The nanopore reads were then aligned back to the polished 16S rRNA gene clusters using vsearch (--id 0.8) (Rognes et al., 2016) to generate an OTU table for subsequent analysis in the QIIME2 v2020.8 pipeline. Taxonomic classification of full length 16S rRNA clusters was via QIIME2 scikit-learn naïve Bayes machine-learning classifier as described above. Similarly, abundance table, taxonomic assignment output, and sample metadata were analysed on the MicrobiomeAnalystCA webserver. Beta diversity profiling and alpha diversity indexes, LefSe and Pearson’s correlation were determined as in the data analysis used to analyse the 16S rRNA V4 region.

Results

General performance of shrimp

Out of the three treatments, two out of three replicates of the control (clearwater) treatment collapsed on day 30. The remaining clearwater treatment had a final survival of 15% and biomass per volume of 1.40 kg/m3. In comparison, RBFT and probiotic treatments had survival rates of 43.7 ± 24.0% and 16.3 ± 9.7%, respectively. Higher biomass per volume was reported in RBFT (2.58 ± 1.20 kg/m3) compared to that of probiotic treatment (0.85 ± 0.49 kg/m3).

Water quality

The biofloc treatments consistently maintained total ammonia nitrogen (TAN) levels below 2 mg/L throughout the entire culture period. Only the control treatments exhibited a higher concentration of 3 mg/L in week 3, which subsequently decreased in week 4 (Fig. 1). However, statistical analysis (P > 0.05, P = 0.155 and H = 3.73, Table 1) revealed no significant difference in TAN concentration among the various treatments. Remarkably, the biofloc treatments successfully maintained TAN levels within a safe range throughout the culture period. Similarly, the concentration of nitrite showed no notable difference between treatments (P > 0.05, P = 0.284 and H = 2.52, Table 1).

Figure 1 Alpha diversity index (observed) of (A) 16S V4 rRNA gene regions using Illumina iSeq 100 and (B) full length 16S rRNA genes using Nanopore sequencing.

Note: Shrimp guts reared in clear water (SIC), probiotic (SIP), and biofloc (SIB); water samples from clear water (CW), probiotic (PW), biofloc of <20 μm (LT20 uM), and biofloc >20 μm (MT20 uM).

Table 1 Total ammonia nitrogen and nitrite measured during 70 days of culture period.

	Total ammonia nitrogen (mg/L)	Nitrite (mg/L)	
Daily water exchange	1.23 ± 0.48	2.03 ± 0.46	
RBFT	0.92 ± 0.30	1.62 ± 0.40	
Commercial probiotic	0.69 ± 0.20	1.52 ± 0.38	
Note:

Daily Water Exchange–control treatment; RBFT and Commercial Probiotic–evaluated treatments. Data presented as mean ± standard error. No significant difference between treatments for total ammonia nitrogen and nitrite.

General statistics of 16S V4 and full length 16S rRNA gene sequencing

Bacterial diversity in water samples, biofloc and shrimp gut were characterised using 16S V4 and full length 16S rRNA gene sequencing via Illumina iSeq10 and Nanopore, respectively. A total of 1,439 and 177 ASVs were identified after processing using the QIIME2 pipeline based on the 16S V4 (average raw read number: 18,864) and full length 16S rRNA (average raw read number: 8,959) gene sequencing, respectively. The plateauing of the alpha diversity rarefaction curves in most samples after 10–20% of sampled read depth for both 16S V4 (Fig. S1A) and full length 16S sequences (Fig. S1B) suggested that the number of reads after QIIME2 processing was sufficient to represent the microbial community diversity for samples in this study. The observed alpha-diversity index was significantly different between treatments based on 16S V4 (ANOVA; F: 16.619, P < 0.001; Fig. 1A) and full length 16S sequences (ANOVA; F: 5.4099, P = 0.021; Fig. 1B) (note: Chao1 results are listed in Table S1). Beta diversity analysis showed that water and gut microbiome samples were similar according to treatments (Fig. 2). Specifically, based on the 16S V4 region, the intestinal bacterial communities of shrimps reared in probiotic and RBFT treatments were similar based on their proximity on the NMDS plot, whereas the intestinal and water sample microbial communities of clearwater treatment formed two small, isolated clusters (Fig. 2A). The water samples of probiotic and biofloc of two sizes (both <20 and >20 μm) showed higher similarity in bacterial communities compared to others. When the full length 16S rRNA gene was analysed, a clearer pattern was observed (Fig. 2B). The intestinal tissues of all treatments (clearwater, probiotic, RBFT) were clearly grouped together. In addition, water samples of clearwater treatment were isolated from probiotic and biofloc of <20 and >20 μm treatments.

Figure 2 Plots of non-metric multidimensional scaling (NMDS) based on Bray-Curtix dissimilarity matrix for (A) 16S V4 rRNA gene region using Illumina iSeq 100 (F: 3.5341, r2: 0.6032, P < 0.001) and (B) full length 16S rRNA genes using N.

Individual samples for each treatment were marked with same colour. Axis values represent variation percentage in the input data that can be explained. Samples in close proximity indicate similarity in microbial composition. Note: Shrimp guts reared in clear water (SIC), probiotic (SIP), and biofloc (SIB); water samples from clear water (CW), probiotic (PW), biofloc of <20 μm (LT20 uM), and biofloc >20 μm (MT20 uM).

Microbial diversity

Microbial diversity varied among treatments regardless of either 16S V4 region (ANOVA; F: 19.050, P < 0.001) or full length 16S (ANOVA; F: 37.992, P < 0.001) were used. Water samples of >20 μm biofloc had the highest observed ASV whereas that of clearwater treatment was among the lowest when based on 16S V4 region. However, when full length 16S was used, no clear distinction was observed in the microbial diversity between water samples and intestine samples (ANOVA; F: 2.396, P: 0.139).

Shannon diversity index differed significantly among treatments (ANOVA; F: 7.912, P: 0.001) based on the 16S V4 region data, with RBFT and probiotic water samples showed significantly higher diversity index compared to the remaining treatments (Fig. 3A). However, although it appeared that microbial diversity in all intestine tissue samples were lower than that of the corresponding water samples based on the full length 16S sequences, the differences were not significant (ANOVA; F: 1.592, P: 0.277) (Fig. 3B). The shared microbiota in combined water and intestine samples based on full length 16S rRNA was approximately five times that of 16S V4 region (Figs. 4A and 4B). When analysed separately, intestine and water samples from biofloc treatment shared more microbiota with that of probiotic treatment compared to control (Figs. 4C–4F).

Figure 3 Shannon diversity index of individual samples (scatterplot) and grouped treatments (boxplot within scatterplot) of (A) 16S V4 rRNA gene regions using Illumina iSeq 100 and (B) full length 16S rRNA genes using Nanopore sequencing. Superscript letters indic.

Note: Shrimp guts reared in clear water (SIC), probiotic (SIP), and biofloc (SIB); water samples from clear water (CW), probiotic (PW), biofloc of <20 μm (LT20 uM), and biofloc >20 μm (MT20 uM).

Figure 4 Venn diagrams showing the unique and shared microbiota in water and intestine samples (A) based on 16S V4 rRNA gene region and (B) based on full length 16S rRNA genes. The unique and shared microbiota of (C) only intestine samples and (D) only water samples based on 16S V4 rRNA gene region, and (E) only intestine samples and (F) only water samples based on full length 16S rRNA genes were further highlighted.

Note: Shrimp guts reared in clear water (SIC), probiotic (SIP), and biofloc (SIB); water samples from clear water (CW), probiotic (PW), biofloc of <20 μm (LT20 uM), and biofloc >20 μm (MT20 uM).

Microbial taxonomic abundance

Based on the 16S V4 and full length 16S sequences, 12 phyla and eight phyla, respectively, could be identified in our samples (Fig. 5), with just five phyla accounting for more than 93% (based on 16S V4 region) and 99% (based on full length 16S sequences) of the sequences obtained. The most abundant taxon was Proteobacteria, representing 59.8% to 86.2% of the total dataset based on abundance using 16S V4 region and 73.3% to 89.7% based on abundance using full length 16S sequences. Based on 16S V4 region, Planctomycetota was among the top four abundant taxa in the water and shrimp intestine samples of probiotic and RBFT treatments, whereas Actinobacteriota was among the top four taxa in water samples (2.9%) and shrimp intestine samples (8.2%) of clearwater treatment. Interestingly, higher abundance percentages of Verrucomicrobiota and Bdellovibrionota were noted in small-sized biofloc water samples (<20 μm) while large-sized biofloc water samples had higher abundance of Planctomycetota and Myxococcota when taxa abundance was analysed based on 16S V4 region. In contrast, when full length 16S was used as the identification baseline, Actinobacteriota and Bacteroidota were among the top three taxa in all treatments, except for shrimp intestine samples subjected to probiotic treatment that was dominated by Cyanobacteria (5.2%) and Actinobacteriota (3.8%) after Proteobacteria (88.3%). Based on 16S V4 region, LDA analysis revealed that Planctomycetota and Chloroflexota were significantly more abundant in biofloc-related samples (biofloc-reared shrimp intestines and biofloc of <20 and >20 μm) whereas water samples but not shrimp intestine samples had significantly more bacteria of phyla Bdellovibrionota and Gemmatimonadota (Fig. 5A).

Figure 5 Proportion of reads assigned at the phylum level in each treatment and dot plot based on (A) 16S V4 rRNA gene region and (B) full length 16S sequences.

Each microbial genus is represented by a different colour in the bar chart. Dot plot of bacterial differential abundance based on full length 16S was not included as results were insignificant as indicated by LefSe analysis (all FDR-adj P > 0.1) (Table S2). Low to high abundance is represented by a change of colour from blue to red in the dot plot. Note: Shrimp guts reared in clear water (SIC), probiotic (SIP), and biofloc (SIB); water samples from clear water (CW), probiotic (PW), biofloc of <20 μm (LT20 uM), and biofloc >20 μm (MT20 uM).

Approximately 13.1% to 36.5% of the ASVs were not assigned to genus level based on the 16S V4 region (Fig. 6A). Among the top genera across samples were Shewanella, Vibrio, Pseudoalteromonas, Rheinheimera, and Photobacterium. Specifically, Vibrio was among the top two genera in the water samples of clearwater and probiotic treatment, with a percentage of 24.0% and 21.5%, respectively. Yet only the intestine tissues of shrimps reared in probiotic treatment exhibited high Vibrio composition percentage (38.2%) whereas that of clearwater treatment had only 9.4%, second to Shewanella (24.6%). Interestingly, the Vibrio composition percentage were low in both <20 (6.6%) and >20 μm (4.9%) RBFT water samples but was the highest genus (28.8%) in the intestine tissues of shrimps subjected to RBFT treatment. The top four dominating bacterial genera were similar in biofloc of <20 and >20 μm, with Pseudoalteromonas being the most prevalent genus. As shown by LDA analysis (Fig. 6A, Table S2), Shewanella was significantly of higher abundance in the shrimp intestinal tissues of clearwater and probiotic treatments, and the water samples of clearwater treatment in comparison to that of biofloc-related samples (Fig. 6A, Table S2). Interestingly, Paracoccus was significantly more abundant in all biofloc-related samples compared to others.

Figure 6 Top 15 proportion of reads assigned at the genus level in each treatment and dot plot (top 10) based on (A) 16S V4 rRNA gene region; Top 15 proportion of reads assigned at the genus level in each treatment and column chart showing the abundance of Vibrio species among treatments based on (B) full length 16S sequences.

Each microbial genus is represented by a different colour in the bar chart. Dot plot of bacterial differential abundance based on full length 16S was not included as results were insignificant as indicated by LefSe analysis (all FDR-adj P > 0.1) (Table S2). Low to high abundance is represented by a change of colour from blue to red in the mini heatmap. No Vibrio was successfully identified to the species level using 16S V4 region. Note: Shrimp guts reared in clear water (SIC), probiotic (SIP), and biofloc (SIB); water samples from clear water (CW), probiotic (PW), biofloc of <20 μm (LT20 uM), and biofloc >20 μm (MT20 uM).

When analysed using full length 16S, Ruegeria dominated (32.5–58.4%) the water and intestine tissues of all treatments (Fig. 6B). Noticeably, all water samples had high percentage of Pararheinheimera (ranging from 8.3% in clearwater treatment to 24.8% in biofloc of <20 μm) but only less than 3.3% in intestinal samples of all treatments. Similar to the results obtained in Fig. 6A, Vibrio was highest in the intestine tissues of shrimps reared in probiotic treatment (4.1%), followed by water samples of clearwater treatment (2.8%) whereas others were ≤1%. However, LDA analysis showed that no bacteria genera show significant difference in abundance between treatments (Table S2).

Vibrio abundance and correlation

Genus Vibrio was successfully characterised to the species level based only on its full length 16S sequences (Fig. 6B). Three Vibrio species were identified, with Vibrio harveyi being the most abundant species in all treatments. In general, V. harveyi, Vibrio parahaemolyticus and Vibrio vulnificus were most abundant in shrimp intestines subjected to probiotic treatment. Among water samples of different treatments, that of clearwater treatment had the most abundant V. harveyi, V. parahaemolyticus and V. vulnificus whereas biofloc water samples of <20 and >20 μm showed the presence of all three Vibrio species but in comparatively lower abundance.

As shown by Pearson’s correlation analysis, there is a positive correlation between Vibrio and Pseudoalteromonas (r = 0.455; P = 0.038), and Simiduia (r = 0.491; P = 0.024) based on both 16S V4 region (Fig. 7A) and between Vibrio and Photobacterium (r = 0.910; P < 0.001) based on full length 16S sequences (Fig. 7B). Subsequently, based on their full length 16S sequences, it is noticeable that the three Vibrio species, i.e., V. parahaemolyticus, V. harveyi, and V. vulnificus, were positively correlated among each other and with other unclassified Vibrio species (Figs. 7C–7E). Also, species showing positive correlations were similar among Vibrio species. For example, Shewanella algae, Photobacterium damselae, and Limnospira fusiformis were positively correlated in V. parahaemolyticus (Fig. 7C), V. harveyi (Fig. 7D), and V. vulnificus (Fig. 7E). Another interesting trend is that most positively correlated taxa within the three Vibrio species were highly abundant in the shrimp intestinal samples of probiotic treatment and water samples of clearwater treatment (Figs. 7C–7E).

Figure 7 Correlation plots between genus Vibrio with other taxa based on (A) 16S V4 region and (B) full length 16S sequencing, and individual Vibrio species, i.e., (C) Vibrio parahaemolyticus, (D) Vibrio harveyi, and (E) Vibrio vulnificus w.

Features are ranked by correlation relationship, with blue bars indicate negative correlations whereas red bars indicate positive correlations. Deeper colouration represent stronger correlation. Low to high abundance is represented by a change of colour from blue to red in the mini heatmap. Note: Shrimp guts reared in clear water (SIC), probiotic (SIP), and biofloc (SIB); water samples from clear water (CW), probiotic (PW), biofloc of <20 μm (LT20 uM), and biofloc >20 μm (MT20 uM).

Size fraction of biofloc

By comparing specifically, the metagenomic profiles between biofloc water samples of <20 and >20 μm, bioflocs of >20 μm harboured lesser Proteobacteria and Verrucomicrobiota but higher percentage of Planctomycetota, Bacteroidota and Actinobacteriota (Figs. 5A and 5B). At the genus level, Pseudoalteromonas and Photobacterium were of higher relative abundance in bioflocs of <20 μm. However, the abundances of V. harveyi and V. parahaemolyticus were similarly low in bioflocs of both sizes, except that V. vulnificus was present in only bioflocs of <20 μm (Figs. 6A and 6B).

Predicted functional profiles of bacterial community in shrimp guts and water samples of clearwater, probiotic and biofloc treatments

To ensure high confidence level, a LDA score of >3 was considered as cut-off value in the prediction of functional profiles of bacterial community from different sources based on the data obtained from 16S rRNA V4 region. A total of 41 significantly enriched functional profiles was found throughout all treatments, except for biofloc of <20 μm (LT20 uM) treatment group (Fig. 8). When cultured in clear water, the bacterial community in shrimp guts (SIC) were enriched in methyl phosphonate degradation and peptidoglycan biosynthesis IV pathways. Gut bacterial community of shrimps subjected to probiotic treatment (SIP) exhibited significant enrichment in 13 biosynthesis pathways, all of which are related to the synthesis of basic molecules that ensure general function of beneficial bacteria such as amine and polyamines, ectoine, 1,4-Dihydroxy-2-naphthoic acid (1,4-DHNA), various enzyme cofactors, and Kdo2-lipid A. Additionally, pathways related to degradation of purine ribo- and deoxyribonucleosides, acetylene, galactose, and chitin derivatives were also enriched in the gut bacterial community of probiotic-treated shrimps. Interestingly, when shrimps were cultured in biofloc treatment (SIB), their bacterial community were enriched in the biosynthesis of enzyme cofactor, fatty acid and lipid biosynthesis, and amino acid biosynthesis, together with the enrichment of functional groups involved in nucleoside and nucleotide degradation, nitrogen compound metabolism, carboxylate degradation, and fermentation.

Figure 8 LDA scores of predicted functional profiles of various treatments.

Functional enrichments with an LDA score of >3 are considered. Note: Shrimp guts reared in clear water (SIC), probiotic (SIP), and biofloc (SIB); water samples from clear water (CW), probiotic (PW), and biofloc >20 μm (MT20 uM).

The bacterial community in the water samples of clear water treatment (CW) were predominantly functionally enriched in the biosynthesis of menaquinol and chorismite metabolism (Fig. 7). In the water samples of probiotic treatment (PW), only the biosynthesis of fatty acid (palmitate) and the superpathway of glycolysis pyruvate dehydrogenase TCA and glyoxylate bypass were significantly enriched. In biofloc >20 μm (MT20 uM), the only two significantly enriched predicted functional profiles were both related to the tricarboxylic acid (TCA) cycle.

Discussion

16S V4 region vs 16S rRNA full length sequencing

Gut microbiota characterisation and data interpretation rely on the selection of specific gene regions, specifically the hypervariable regions of the 16S rRNA gene, such as V3, V4, V5, either alone or in combination (Garcia-López et al., 2020; Fadeev et al., 2021). Hypervariable regions of V3-V4 are commonly employed to investigate gut microbial diversity in P. vannamei (Zoqratt et al., 2018; Fan & Li, 2019). Although other regions, such as V2, V3, V4, V6-V7, V8 and V9 have been used to characterise the structure and function of gut microbiota of shrimps as well (Cornejo-Granados et al., 2017). One limiting factor during the selection of 16S rRNA hypervariable region is cost effectiveness–longer sequences translate to higher sequencing cost. Thus, researchers have started to resort to only using one hypervariable region, i.e., V4 region, to analyse the internal microbiota of penaeid shrimps (Cardona et al., 2016; Imaizumi et al., 2021). In addition, Onywera & Meiring (2020) showed that the microbiota profiles from one specific hypervariable region (i.e., V3) and the longer V3-V4 regions are comparable. Targeting shorter fragments of the 16S rRNA, albeit its cost effectiveness due to shorter sequences, however, is vulnerable to identification bias owing to the potential production of chimeric sequences, and possess lesser identification power (commonly up to genus level), limiting to only strains with high similarity that can be identified to the species level (Shin et al., 2016). Recently, the availability to sequence full length 16S rRNA sequences at an affordable rate enables researchers another feasible alternative to understand bacterial composition and functional changes in specific condition. Specifically, the pipeline for full length 16S rRNA sequencing using nanopore long-read analyzer is mature and is being used by researchers for various microbial studies (Matsuo et al., 2021; Huggins et al., 2022; Matsuo, 2023). Full length 16S rRNA allow species-level identification (Matsuo et al., 2021) and this is highlighted in our study as well, where Vibrio can only be identified to the species level using 16S rRNA sequencing. This study detail the first employment of nanopore-based sequencing to obtain full length 16S rRNA sequences from shrimps and water samples in the aquaculture sector.

Microbial structure and composition

The intestinal microbial community of shrimps are affected by the microbes present within the water column and ingested particles, including bioflocs. Bioflocs contains a diverse niche of microbes are known to confer positive impacts, especially improved innate immunity towards cultured shrimps by altering the bacterial composition in the shrimps’ intestinal microbiome (Dauda, 2020; Yee et al., 2021). Yet, the shift in water and intestine microflora following probiotic and BFT administration is highly dependent on the active bacteria present (Dauda, 2020). As shown in the consistency of clustering patterns using both 16S V4 region and full 16S sequence, the microbial community of water and shrimp intestines subjected to additional introduction of beneficial bacteria in the form of probiotic and RBFT were similar while that of clearwater treatment was distinctively separated. Similar distinct in microbial community dissimilarity between BFT or probiotic and clearwater was also reported in other studies on penaeid shrimps (de Souza Valente et al., 2020; Tepaamorndech et al., 2020). Clear separation between water and intestinal microflora in all treatments were expected due to the diversity of bacteria found within shrimp’s intestine compared to water column (Deng et al., 2019).

The dominancy of Proteobacteria found across treatments is consistent in other P. vannamei studies (Fan & Li, 2019; Wang et al., 2020a) and other crustacean species, including tiger shrimp Penaeus monodon (Angthong et al., 2020), Chinese mitten crab Eriocheir sinensis (Ding et al., 2017), and mud crab Scylla paramamosain (Wei et al., 2019). Proteobacteria is made up of gram-negative bacteria and this phylum is highly diverse (Holt et al., 2020). Their dominancy, however, can be affected by various factors, including diseases. For example, shrimps with white faeces syndrome had lower Paracoccus (Proteobacteria) and Lactococcus (Firmicutes) but an increase in Phascolarctobacterium (Firmicutes) and Candidatus (Tenericutes) (Hou et al., 2018).

Vibrios

The abundance of Vibrio among treatments is interesting, in which among water treatments, that of RBFT had the lowest Vibrio abundance whereas that of probiotic and clearwater were high; among shrimp intestines, Vibrio abundance was lowest in clearwater treatment but high in RBFT and probiotic treatments. Bacteria of the genus Vibrio are widely distributed in the marine environment and are often found as part of the microflora in shrimp culture systems (de Souza Valente & Wan, 2021). The significant reduction in Vibrio abundance in RBFT is expected as bioflocs are known to produce inhibiting compounds with antibacterial properties. For instances, biofloc microorganisms produce poly-b-hydroxybutyrate (PHB), a preventive and curative compound against Vibrio infections (El-Sayed, 2021). Vibrio, being a commonly found genus of the natural gut microflora of healthy P. vannamei (Wang et al., 2020a), are often non-pathogenic and could even be beneficial with probiotic potential (Ninawe & Selvin, 2009; Rajeev et al., 2021). For example, V. alginolyticus isolated from the gastrointestinal tract of adult P. vannamei serves as a probiotic against V. parahaemolyticus (Balcázar, Rojas-Luna & Cunningham, 2007). Also, many Vibrio species are involved in the production of chitinolytic enzymes and are being postulated to be one of the reasons of its abundance in the chitin-rich crustacean gut (Holt et al., 2020). Therefore, based on the results of this study, it is postulated that RBFT reduced pathogenic Vibrio spp. in the water column but increases the abundance of beneficial Vibrio spp. within the gut microflora of P. vannamei. Future research on the detailed identification of Vibrio species would aid in supporting this postulate.

Further, only three Vibrio species were successfully identified to species level based on the full length bacterial 16S sequences. The beneficial effect of probiotic (Bacillus spp.) and RBFT was clearly seen in the water samples after 70 days by the lower abundance of V. harveyi, V. parahaemolyticus, and V. vulnificus compared to clearwater treatment. Interestingly, the gut microflora of probiotic treatment was the highest in the abundance of V. harveyi, V. parahaemolyticus, and V. vulnificus, while that of RBFT were the lowest in terms of V. harveyi abundance. Vibrio harveyi, a known pathogen in fish and crustaceans, and the etiological agent of luminous vibriosis in shrimp, has been reported to cause mass mortalities in P. vannamei (Zhou et al., 2012). Vibrio parahaemolyticus is the causative agent of acute hepatopancreas necrosis disease (AHPND) in shrimp (Aguilar-Rendón et al., 2020) while together with V. parahaemolyticus and V. cholera, V. vulnificus is considered as one of the three majors pathogenic Vibrios that causes serious human health concerns, including V. vulnificus sepsis (Teng et al., 2017). The higher abundance of all three Vibrio species in the shrimp intestines subjected to probiotic treatment might reflect the unsuitability of Bacillus spp. against these specific pathogens. Some Bacillus species, such as B. subtilis BS11 and Bacillus sp. P11 showed no extracellular antimicrobial activity (Powedchagun, Suzuki & Rengpipat, 2011; Utiswannakul, Sangchai & Engpipat, 2011). Probiotic B. subtilis E20 from the gut microflora of P. vannamei, although resulted in higher shrimp survival after administration, had no inhibitory effects against common shrimp pathogens including V. vulnificus and V. alginolyticus. Significant increase, however, was observed in several immune parameters including phenoloxidase activity, phagocytic activity, and clearance efficiency (Tseng et al., 2009). Similarly, as shown by the higher shrimp survival and growth in our study, probiotic and RBFT might increase shrimp resistance towards pathogen by indirect routes of immune modifications as in the case of B. subtilis E20 (Tseng et al., 2009). Further study on the characterisation of immune response in shrimps would aid in supporting this postulate. In addition, BFT has been proven to enhance P. vannamei survival by inducing V. parahaemolyticus AHPND strain to switch from free-living virulent planktonic phenotype to non-virulent biofilm phenotype (Kumar et al., 2020). This would explain the slightly higher V. parahaemolyticus abundance in the shrimp intestines of RBFT treatment than that of clearwater, but also with better final output of shrimps.

Correlations between the three Vibrio species, i.e., V. parahaemolyticus, V. harveyi, and V. vulnificus, and between Vibrio and Photobacterium, as indicated in our study, highlight the potential connectivity between pathogenic bacteria. By using molecular ecological networks and null community modelling, Huang et al. (2020b) pointed out that drift process of interspecies interaction between bacterial communities increased in diseased shrimps. Thus, shrimps that are immune compromised by a single pathogenic agent would be of higher risk to the overgrowth of other opportunistic pathogens (Soto-Rodriguez et al., 2015). Furthermore, the strong correlations found between Vibrio and Photobacterium, and between Vibrio species might imply that there is drift in bacterial communities when particular pathogen proliferates and subsequently resulting in the proliferation of other opportunistic pathogens. The lower V. harveyi and overall pathogenic Vibrio abundance in RBFT compared to probiotic treatment, coupled with the low Vibrio abundance in the water column of RBFT treatment compared to probiotic and control highlight the beneficial impact of biofloc over probiotic.

In addition, probiotics can be added to a biofloc system in shrimp culture to potentially improve shrimp production, although the selection probiotics is crucial. The addition of commercial probiotics did not improve the growth, survival and feed conversion rate of whiteleg shrimp during the nursery phase (Arias-Moscoso et al., 2018) whereas Amjad et al. (2022) reported that the probiotics-added biofloc systems improved water quality and growth of P. vannamei juvveniles.

Size fraction of biofloc

BFT often results in microbial bioflocs of various sizes with varying nutritional composition (Ekasari et al., 2014). In this study, we showed that large-sized bioflocs (>20 μm) of RBFT exhibited different bacteria composition relative abundance compared to small-sized bioflocs of <20 μm. Particle-attached bacteria are more prevalent in large-size bioflocs whereas small-sized bioflocs would harbour more free-living bacteria, as evident in the study of Huang et al. (2020a). Planctomycetota and bacteroidota exhibit degradative capabilities and often adhere to the surface of invertebrates, detrital aggregates, and phytoplankton (Lage & Bondoso, 2014; Huang et al., 2020a). The larger surface area of bioflocs >20 μm and the preference of bacteria of these phyla would explain the higher abundance of Planctomycetota and bacteroidota in large-sized bioflocs (>20 μm).

Bioflocs of <20 μm harboured higher abundance of Pseudoalteromonas and Photobacterium. The genus Pseudoalteromonas is commonly found in marine environment and contains both beneficial probiotic species and known pathogenic species towards shrimps (Tzuc et al., 2014; Wang et al., 2018; Alfiansah et al., 2020). In particular, together with Vibrio, Pseudoalteromonas and Photobacterium are among the potential pathogenic gastrointestinal taxa that are linked with the occurrence of white feces disease in shrimp (Huang et al., 2020b). Also, all three Vibrios (V. harveyi, V. parahaemolyticus, and V. vulnificus) were present in small-sized bioflocs whereas V. vulnificus was absent in large-sized bioflocs. Therefore, concurred to the findings of Huang et al. (2020a), this study also noticed that small-sized bioflocs contained higher abundance of potential pathogens that might increase the chances of contracting diseases in cultured shrimps.

Predicted functional profiles of bacterial community

Predicting functional profiles from 16S rRNA data provides an overview of the functional potential of the microbial community and is regarded as an indispensable tool to study the modulatory effect of bacteria on cellular and molecular level of host (Ortiz-Estrada et al., 2019; Foysal, Momtaz & Kawser, 2021; Mongad et al., 2021). Metagenomic functional analysis was conducted using PICRUSt2 (Douglas et al., 2020), a well-known pipeline for the prediction of gene functions based on 16S rRNA data and is widely used in hosts of aquaculture importance, including cultured fish and shrimp (Foysal, Momtaz & Kawser, 2021; Amillano-Cisneros et al., 2022; Zhao et al., 2022).

The upregulation of methyl phosphonate degradation pathway and peptidoglycan biosynthesis IV pathway in the shrimp gut bacterial community of clear water treatment is expected because methyl phosphonate degradation is a common pathway in which bacteria utilise methyl phosphonate found in the gut of the host as the sole source of phosphorus whereas the latter is the synthesis of peptidoglycan by bacteria under the order Lactobacillales (known taxa that possess this pathway includes Enterococcus faecium, Enterococcus hirae, Lactococcus lactis lactis), an important constituent of cell walls of almost all eubacteria. A recent study by Kim et al. (2019) further shows that E. faecium and its secreted peptidoglycan hydrolase (SagA) could improve intestinal barrier function and protect host from enteric pathogens. Subjecting shrimps to probiotic treatment clearly led to the upregulation of various biosynthesis and degradation pathways that have crucial role in digestion and immunity of host by the gut microbial community, reflecting the known benefits of probiotic bacteria (Sumon et al., 2022; Wei et al., 2022). The introduction of biofloc in shrimp culture yielded positive results as it steered shrimp gut microbiota towards the upregulation of fatty acid and amino acid biosynthesis pathways, thereby enhancing host nutrient metabolism and growth (Khanjani & Sharifinia, 2020; Zablon et al., 2022). As one of the major components of biofloc is nitrifying bacteria (Souza et al., 2019), pathways related to fermentation and urea cycle were also enriched in gut microbial community of shrimps exposed to biofloc.

Specifically comparing the metabolic pathway changes in shrimp guts subjected to biofloc and probiotics, that of probiotic-treated shrimp guts bacterial communities exhibited significant metabolic profiles related to the general function of beneficial bacteria whereas that of biofloc-treated shrimp guts bacterial communities encompassed pathways related to nutrient biosynthesis and degradation, urea cycle and fermentation. The involvement of biofloc microbiota in providing enzymes that work in synergy with endogenous host enzymes to improve nutrient digestion and assimilation is also reported in other aquatic species (Gullian-Klanian, Quintanilla-Mena & Hau, 2023). Additionally, only gut microbiota in biofloc treatment showed involvement in urea cycle pathway as expected, because urea is the main nitrogenous source in the process of biofloc formation (Nisar et al., 2022). Thus, the introduction of biofloc in the water column also altered the gut microbiota of cultured organisms, promoting the proliferation of microbiota capable of metabolising nitrogenous compounds, thereby highlighting its added advantage over the usage of only probiotics in shrimp culture.

Pathways upregulated in the bacterial community within water column in the control group (clear water treatment) have important role in bacterial metabolism and overall cell fitness because menaquinones are involved in the regulation of membrane fluidity (Flegler, Kombeitz & Lipski, 2021), mediation of electron transport between various enzymes (van Beilen & Hellingwerf, 2016; Boersch et al., 2018), and the generation of ATP (Boersch et al., 2018). Additionally, menaquinones serve as crucial source of essential vitamin (vitamin K2) for animals, where vitamin K2 is needed various physiological processes, including in blood coagulation (Walther et al., 2013). Probiotic introduction induced upregulation of nutrient (glucose and fatty acid) metabolism pathways in the water column, thereby removing organic matter load and ensuring better water quality (Hlordzi et al., 2020). Other benefits of probiotic inclusion into the water column of aquaculture species includes improve nutrient digestibility, increase stress tolerance, and enhance reproduction (Liu et al., 2010; Toledo et al., 2019; Wang et al., 2020b). Similarly, the introduction of biofloc into the water column also aided in the catabolism of organic fuel molecules via the upregulation of TCA cycle pathways by heterotrophic bacteria in biofloc (Abakari, Luo & Kombat, 2021).

Conclusion

The microbial structure and composition among treatments in this study is consistent with other reported studies, and with clear separation between water and intestinal microflora in all treatments. Water column harboured lower number of Vibrios when biofloc was applied; intestinal Vibrios, however, were high in both probiotic and biofloc treatments, presumably due to probiotic incompatibility and the ability of biofloc to convert virulent to non-virulent Vibrios. Beneficial bacteria such as those of Planctomycetota and bacteroidota were more abundantly found in large-sized bioflocs whereas small-sized bioflocs contained higher abundance of potential pathogenic bacteria such as Pseudoalteromonas and Photobacterium.

Supplemental Information

Supplemental Information 1 Chao1 data of V4 and full length 16S sequencing.

Click here for additional data file.

Supplemental Information 2 LefSe analysis based on phylum and genus using 16S V4 region and full length 16S sequences.

Click here for additional data file.

Supplemental Information 3 Pearson’s correlation analysis of Vibrio genus and individual Vibrio species.

Click here for additional data file.

Supplemental Information 4 Alpha diversity rarefaction curves for (A) 16S V4 rRNA gene regions using Illumina iSeq 100 and (B) full length 16S rRNA genes using Nanopore sequencing. Each treatment had three and two replicates, respectively. Curves started to plateau after reaching 1.

(A) shrimp guts reared in clear water (1CW1-SIC, 1CW2-SIC, 1CW3-SIC), probiotic (1P11-SIP, 1P12-SIP, 1P13-SIP), and biofloc (IB8-SIB, IB9-SIB, IB10-SIB); water samples from clear water (CW1-CW, CW2-CW, CW3-CW), probiotic (PB11-PW, PB12-PW, PB13-PW), biofloc of <20 μm (T8-LT20 uM, T9-LT20 uM, T10-LT20 uM), and biofloc >20 μm (T8-MT20 uM, T9-MT20 uM, T10-MT20 uM). (B) shrimp guts reared in clear water (ICW1, ICW2), probiotic (IP11, IP12), and biofloc (IB8, IB9); water samples from clear water (CW1, CW2), probiotic (PB11, PB12), biofloc of <20 μm (T8LT20, T9LT20), and biofloc >20 μm (T8MT20, T9MT20).

Click here for additional data file.

Additional Information and Declarations

Competing Interests

Author Contributions

Patent Disclosures

Data Availability

Khor Waiho is an Academic Editor for PeerJ. Muhammad Syafiq Abd Razak, Mohd Zaidy Abdul Rahman, and Zainah Zaid are employees of Zaiyadal Aquaculture Sdn. Bhd.

Khor Waiho conceived and designed the experiments, analyzed the data, authored or reviewed drafts of the article, and approved the final draft.

Muhammad Syafiq Abd Razak performed the experiments, analyzed the data, prepared figures and/or tables, and approved the final draft.

Mohd Zaidy Abdul Rahman performed the experiments, prepared figures and/or tables, and approved the final draft.

Zainah Zaid performed the experiments, prepared figures and/or tables, and approved the final draft.

Mhd Ikhwanuddin performed the experiments, prepared figures and/or tables, and approved the final draft.

Hanafiah Fazhan performed the experiments, prepared figures and/or tables, and approved the final draft.

Alexander Chong Shu-Chien analyzed the data, authored or reviewed drafts of the article, and approved the final draft.

Nyok-Sean Lau analyzed the data, authored or reviewed drafts of the article, and approved the final draft.

Ghazali Azmie performed the experiments, analyzed the data, authored or reviewed drafts of the article, and approved the final draft.

Ahmad Najmi Ishak performed the experiments, authored or reviewed drafts of the article, and approved the final draft.

Mohammad Syahnon performed the experiments, authored or reviewed drafts of the article, and approved the final draft.

Nor Azman Kasan conceived and designed the experiments, authored or reviewed drafts of the article, and approved the final draft.

The following patent dependencies were disclosed by the authors:

RBFT is patented under Patent ID: PI 2017703679; Trademark ID: TM2021014913.

The following information was supplied regarding data availability:

The sequences are available at GenBank: PRJNA736602 (16S V4 region mitogenome sequencing) and PRJNA736625 (full-length 16S mitogenome sequencing).

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
