# Peer review of "A metagenomic comparison of clearwater, probiotic, and Rapid BFTTM on Pacific whiteleg shrimp, Litopenaeus vannamei cultures"

_PeerJ, doi:10.7717/peerj.15758_

## Round 0.1 · original submission · Minor Revisions

Your manuscript needs to be improved. More concretely, better work on the figures and a deep analysis of them will improve the results and will enrich the discussion section. In other words, the manuscript will be more readable and attractive to the audience.

Reviewer 1 ·

Basic reporting

The report is clear, and the data support the conclusions. The grammar is adequate, although the style may be improved for better understanding.
The inclusion of Venn diagrams and tables also may help in understanding and comparison of treatments.
It is also recommended to review a paper on the effect of BioFloc on the culture of L. stylirostris (Cardona et al. BMC Microbiology :(2016)16157 DOI 10.1186/s12866-016-0770-z) since they also used only V4 for molecular taxonomy. They present in a table in an understandable way the changes in the relative abundance of the bacterial phyla comparing marine water vs. Biofloc. Also, Venn diagrams are useful to show the core microbiome and the changes by treatment and organ.

Experimental design

16S rRNA V4 for taxonomical identification is not used in most shrimp metagenomics reports. V3-V4 is most commonly used (Garcia-Lopez et al. 2020. Microorganisms 8(1):134 https://doi.org/10.3390/microorganisms8010134). V4 alone is less informative, and using the whole ribosomal region sequenced by Nanopore gives different results.
Please explain why using both strategies instead of focusing on only one region and the nanopore sequencing.

Validity of the findings

The results where Biofloc provides the best shrimp production yield were expected. However, the survivals were much lower than reported in the literature (Xu et al., 2016. Aquaculture
453, pp. 169-175). Please report more physicochemical variables for the water in the three treatments to explain the results possibly.
Please discuss the methodological aspects and compare the V4 vs. Nanopore full 16S sequencing. Which is better regarding cost and information obtained?
Also, the discussion on the metabolic pathways could be improved regarding the metabolic contribution comparing the whole BioFloc vs. simple probiotics.

Additional comments

The paper may improve by emphasizing a few messages:

Is BioFloc suitable for shrimp production? Costs and benefits vs. simple probiotics
The survival rates obtained are typical in Malaysia, or what are the reasons for those results?
The methodological comparison between V4 and nanopore 16S sequencing.

Reviewer 2 ·

Basic reporting

The manuscript by Waiho et al. characterizes the metagenome derived from water and shrimp intestine samples of Rapid BFT with probiotic and Clearwater treatments using 16s rRNA V4 and full-length Regions. It is a concise manuscript regarding the metagenome analysis of current aquaculture technology.
However, a few minor details should be addressed before accepting the manuscript for publication.

Experimental design

The manuscript has a well-defined question with an adequate experimental design. Therefore, I only have a few concerns described in the additional comments.

Validity of the findings

Overall the statistical analysis and presentation of the results seem adequate and sufficient to answer the main question and support the conclusions. I have a few concerns about the figures, which I describe in the additional comments.

Additional comments

1. In line 188, the authors specify using ASVs for the V4 region sequence analysis. However, across the manuscript, they also mention “OTUs table,” “estimated OTU,” “observed OTU,” etc. Please rectify these terms, as they may be confusing for the reader. Consider replacing OTUs for features when necessary.
2. Specify the quality filters used to clean the sequencing data of the 16S V4 region.
3. Consider adding ellipses to the clusters in Figure 2. They should help to make the Figure more clear. You can add the ellipses with a simple script in R.
4. Consider dividing Figure 3 into two Figures—the first including the boxplots and a supplementary figure with the data of all samples.
5. To better understand the saturation or “plateau” reached in the alpha diversity analysis, consider expressing Figure 1 in “observed features” instead of the PD diversity index and add the value of the Chao1 index in the results section.
6. In line 333, the Pearson correlation coefficient is expressed with “r” instead of “P.” Please correct.
7. Rectify the < symbol in P<0.5. I believe it should be >0.5. Also, stating that the correlation is “highly positively” can be misleading. I suggest just indicating there is a positive correlation.
8. Add a color scale in the legend of Figure 6 to better understand the heatmaps.
9. Please indicate the normalization method used to express the taxa abundance in Figures 4 and 5. The figure legend states “proportion of reads”; does that means you did not consider a minimum read abundance to dismiss any transitional bacteria?
10. The discussion feels a little simple. Consider adding insight on how this study supports or discourages using Biofloc; and what were the advantages of sequencing the 16S rRNA full length.
11. Please revise the overall writing of the manuscript. There are many typos across the document. For example:
Line 123 “…biofloc sedimentable solid level is…” should be “was”
Line 325 says, “based only based”
Line 332 says “…comparatiely low abundance…” should be “lower”
Data availability says, “…region mitogenome sequencing and PRJNA736625 for full0length 16S mitogenome sequencing.”
And so on. Please correct accordingly.
12. Out of curiosity, what was the abundance of the probiotic genus (Bacillus spp.) after the treatment?

---

## Round 0.2 · accepted · Accept

Thank you for submitting your work to this journal.

With kind regards,

Reviewer 1 ·

Basic reporting

The quality of the manuscript has improved significantly, making it more readable and easier to follow the discussion on comparing V4 and Nanopore-full 16S sequencing. Additionally, the detailed information on sample preparation for sequencing is valuable for readers who wish to reproduce the experiments.

Experimental design

Thank you for providing additional details on the experimental design and sequencing data analysis. Your efforts to enhance the Venn diagrams and other diagrams are also greatly valued.

Validity of the findings

Analyzing the statistics of the data is crucial, as is comparing the results from Nanopore 16S and iLumina V4 sequencing.

Additional comments

Great job! The paper has shown significant improvement.